# Analysis of Lightweight Cryptographic Algorithms on IoT Hardware Platform †

**Mohammed El-hajj** [1,*,‡], **Hussien Mousawi** [2,‡] and **Ahmad Fadlallah** [3,‡]

1 Faculty of Electrical Engineering, Mathematics and Computer Science, University of Twente, 7500 AE Enschede, The Netherlands
2 Faculty of Engineering, Lebanese University, Beirut 1533, Lebanon
3 Faculty of Arts and Science, University of Sciences and Arts in Lebanon, Beirut 1002, Lebanon
* Correspondence: m.elhajj@utwente.nl
† This paper is an extended version of the conference paper.
‡ These authors contributed equally to this work.

**Abstract:** Highly constrained devices that are interconnected and interact to complete a task are being used in a diverse range of new fields. The Internet of Things (IoT), cyber-physical systems, distributed control systems, vehicular systems, wireless sensor networks, tele-medicine, and the smart grid are a few examples of these fields. In any of these contexts, security and privacy might be essential aspects. Research on secure communication in Internet of Things (IoT) networks is a highly contested topic. One method for ensuring secure data transmission is cryptography. Because IoT devices have limited resources, such as power, memory, and batteries, IoT networks have boosted the term "lightweight cryptography". Algorithms for lightweight cryptography are designed to efficiently protect data while using minimal resources. In this research, we evaluated and benchmarked lightweight symmetric ciphers for resource-constrained devices. The evaluation is performed using two widely used platform: Arduino and Raspberry Pi. In the first part, we implemented 39 block ciphers on an ATMEGA328p microcontroller and analyzed them in the terms of speed, cost, and energy efficiency during encryption and decryption for different block and key sizes. In the second part, the 2nd-round NIST candidates (80 stream and block cipher algorithms) were added to the first-part ciphers in a comprehensive analysis for equivalent block and key sizes in the terms of latency and energy efficiency.

**Keywords:** IoT; constrained devices; LWC; lightweight cryptography; Raspberry Pi; Arduino

## 1. Introduction

The Internet of Things (IoT) security is a strongly contested research topic. The IoT is a type of network paradigm that uses sensor and Internet technology to transform everyday items into smart devices [1]. Such devices give people the ability to be connected anytime, anywhere, using any connectivity to benefit from a wide spectrum of services [2]. So, digitization is not an option anymore, where it is involved in our daily life activities, including smart homes, smart cities, wearables, e-health, etc. [3]. The IoT end-devices are often operating in vulnerable environments, which leads to several security challenges that should be taken into consideration [4]. To overcome such challenges, various researchers have developed different cryptographic algorithms that can be used to secure IoT applications in order to ensure data protection and data privacy. However, traditional cryptographic algorithms are not suitable to be implemented in the resource-constrained devices used in such an application. The concept lightweight cryptographic (LWC) schemes arose to reflect the need of cryptographic algorithms that provide security with the use of an efficient amount of resources [5]. This resource usage is determined by the key size, the number of rounds, the block size, the memory usage (ROM and RAM), the structure, and the execution time. The objective of the lightweight algorithms' creation is to strike a balance in

several aspects, such as performance, low resource demand, and cryptographic algorithm strength and stability [6]. Various LWC algorithms were proposed as replacements for the current standards, the Advanced Encryption Standard (AES0) [7], Rivest–Shamir–Adleman (RSA) [8], etc., such as TWINE [9], PRESENT [10], SIMON and SPECK [11], QARMA [12], block ciphers, hash functions, and the stream cipher, to enforce the security. In spite of the expanding request in this new area of research, few research works presented the benchmarking and comparison of the well-known LWC algorithms between different hardware platforms of constrained devices. Moreover, no article presented nor involved the software implementation with an analysis and a comparison of any lightweight cryptography on Raspberry Pi compared to others. The objective of this research work is to provide a comprehensive benchmarking of well-known lightweight cryptographic algorithms. These benchmarking results are obtained by a software implementation of the chosen algorithms, implemented on the microcontroller ATMEGA328P-Arduino (Uno) and the Raspberry Pi. To the best of our knowledge and based on the literature review, this work is considered the first to evaluate the performances of lightweight cryptographic schemes. A total of 119 different schemes were evaluated.

This project's main contribution was to evaluate and benchmark lightweight symmetric ciphers for resource-constrained devices. The evaluation is performed using two widely used platform: Arduino and Raspberry Pi. In the first part, we implemented 39 block ciphers on an ATMEGA328p microcontroller using the Arduino platform and on the Raspberry Pi. The block cipher implementations were analyzed in terms of the speed, cost, and energy efficiency during encryption and decryption for different block and key sizes. In the second part, the 2nd-round NIST candidates (80 stream and block cipher algorithms) were added to the first-part ciphers in a comprehensive analysis for equivalent block and key sizes of latency and energy efficiency in encryption and decryption using the two boards. Further, in this part, a method of referencing and comparing is reckoned that adds more of an estimation approach of analysis. The motivation for analyzing the performance of lightweight cryptography algorithms is to identify which algorithms are the most efficient and secure for use in resource-constrained devices, such as IoT devices, mobile devices, and embedded systems. This allows for the selection of the best algorithm for a given application and for the optimization of the implementation of the chosen algorithm to minimize resource usage and increase security. Additionally, it also helps in identifying any weaknesses or vulnerabilities in the algorithm that could be exploited by attackers. The rest of this paper is structured as follows: Section 2 provides a literature review about the work conducted related to the implementation of an LWC using different hardware platforms. Section 3 details the software and hardware setup performed, the measuring metrics used, and the methodology applied to evaluate the communication and computation cost of implementing such schemes. Section 4 presents the results of the experimentation, and Section 5 analyzes and discusses the results achieved. Finally, Section 6 concludes the paper and presents areas of interest for further research.

## 2. Background

The research and development of lightweight cryptography for use on resource-constrained IoT devices have advanced quickly over the past decade. The main goal is to create and use simple cryptographic algorithms that may be applied to such applications while providing the appropriate levels of security. The implementations for IoT applications can be categorized as either software or hardware solutions. For the hardware implementation, the speed, the area, and the energy consumption are taken into consideration. For the software implementation, the required memory size (ROM and RAM) of the embedded software is taken into consideration. When selecting the proper security algorithm to be utilized for resource-constrained devices, these restrictions must be respected [13].

With a focus on LED [14], Piccolo [15], and PRESENT [16], the authors in [17] investigated various software implementations of lightweight ciphers for x86 processors. First, they examined table-based implementations and then offered a theoretical model to

forecast how different potential trade-offs will behave in relation to the processor cache delay profile.

For IoT platforms, such as resource-constrained devices (8-bit AVR and 32-bit ARM Cortex-M3) and Application-Specific Integrated Circuits, the authors in [18] studied the lightweight properties of the HIGHT [19] block cipher and offered the optimized implementations of both software and hardware for IoT platforms, such as resource-constrained devices (8-bit AVR and 32-bit ARM Cortex-M3) and Application-Specific Integrated Circuits.

Utilizing a unique, lightweight reconfigurable processor, the authors in [20] enabled this comparison in their study. Six ciphers including AES, SIMON, SPECK, PRESENT, LED, and TWINE were implemented in the hardware with a Register-Transfer-Level (RTL) design [21] and in the software with a specially designed reconfigurable processor. A direct comparison of the area, throughput, power, energy, and throughput-to-area (TP/A) ratio was performed. Both hardware and software versions were implemented in an identical Xilinx Kintex-7 FPGAs SIMON [22], a lightweight block cipher designed for hardware implementation. Implementing, optimizing, and modeling the SIMON cipher design for resource-constrained devices with a focus on energy and power were the goals of the research conducted by the authors in [23]. The scalar and pipelines design implementations FPGA technology were the two types that were explored in this research [24].

The hardware implementation of the block cipher RECTANGLE with various data paths was the focus of the authors in [25]. They devised, constructed, and assessed the five most effective RECTANGLE [26] cipher data paths for various data bus sizes. The same implementation conditions were used for all of these data paths when they were implemented on various FPGA platforms, and the results were compared across all performance metrics. The ideal architecture for an application can be chosen based on the device and desired performance metrics.

Meanwhile, in [27], the authors used Artix-7, Spartan-6, and Cyclone-V FPGAs to implement the six NIST LWC round 2 candidate ciphers, SpoC, GIFT-COFB [28], COMET-AES [29], COMET-CHAM [29], Ascon [30], and Schwaemm and Esch [31]. Among all the schemes, it was clear that SpoC had the lowest area and power consumption, while Ascon had the highest throughput-to-area (TPA) ratio. KLEIN-80, TWINE-80, Piccolo-80, SPECK (64, 96), and SIMON (64, 96) were among the choices of lightweight block ciphers that were implemented on the Atmega128 processor in the AVR studio 5.1 simulation environment performed by the authors in [32]. The evaluation's findings indicate that the SPECK (64,96) cipher was the most energy efficient and is suitable for wireless sensor networks. Meanwhile, the implementation of the TWINE-80 was the most appropriate with regard to memory utilization.

The authors in [32] studied the efficiency of lightweight block ciphers when used in resource-constrained environments. Specifically, the focus was on the "Saturnin" family of ciphers. The study analyzed how well Saturnin performs when implemented in a specific resource-constrained environment. In addition, to evaluate the results, a comparison with the Advanced Encryption Standard (AES) was conducted using an experimental setup. The findings showed that substantial performance improvements can be achieved as Saturnin, which is based on the design of the AES, can be almost twice as fast as the AES in such restricted environments.

In [33], the authors aimed to present an overview of the current state of the lightweight cryptography algorithms used in IoT environments. A comprehensive review of the literature was conducted to identify different algorithms and extract relevant data, such as the level of security, encryption and decryption performance, execution time, memory usage, clock speed, latency, and frequency. The data were then presented in comparison tables for a further analysis, evaluation, and assessment. The study provided insight into the performance of the lightweight cryptography algorithms used in IoT environments and devices. Additionally, it suggested future research directions to build on the findings of the study.

Almost all the cited works were interested in the implementation of specific LWC schemes, while in this work we performed a software implementation of almost 119 different schemes and compared their performance using two different hardware platforms.

## 3. Method and Experimental Setup

Software implementation is the targeted initiation of comparison between microcontroller and microprocessor platforms. The environment for such comparative analysis is established upon a package of chosen algorithms and a batch of realized metrics entities for measurements. However, these metrics are generalized as sizes of memory variety, speed, throughput, and latency. Beyond, the energy measurement concept is considered additionally for advanced comparison analysis.

### 3.1. Algorithms Used for Evaluation

Lightweight cryptography of various structures, key sizes, and block sizes were chosen. A wide range of differences in key size, block size, and rounds were realized as essential for analysis goals. A total of 39 different ciphers of 13 families shown in Table 1. Furthermore, a package of 80 algorithms of 32 families presented in NIST round 2 competition [34] are included in an extended study shown in Table 2. In Table 1, a rundown of the elected lightweight block ciphers is provided, arranged in alphabetical order, with their type, structure, block size in bits, key size in bits, and the number of rounds. NIST 2nd-round candidates in competition are presented in Table 2, with family type, block size in bits, key size in bits, the number of rounds, and some brief information. These candidates are expected to be selected as a replacement of the traditional standards.

**Table 1.** Thirty-nine algorithms and AES as a relative reference.

| Family-Cipher | Algorithm | Type | Structure | Block Size (bits) | Key Size (bits) | Rounds |
|---|---|---|---|---|---|---|
| Relative reference | AES | Block Cipher | SPN | 128 | 128/192/256 | 10/12/14 |
| 1-1 | HIGHT | Block Cipher | Generalized Feistel Structure (GFS) | 64 | 128 | 32 |
| 2-2/3/4 | KATAN | Block Cipher | stream cipher like | 32/48/64 | 80 | 254 |
| 3-5/6/7 | KTANTAN | Block Cipher | Stream cipher like | 32/48/64 | 80 | 254 |
| 4-8/9/10 | LEA | Block Cipher | Generalized Feistel Network (GFN) | 128 | 128/192/256 | 24/28/32 |
| 5-11 | Piccolo | Block Cipher | GFN | 64 | 80/128 only 80 chosen | 25/31 |
| 6-12/13 | PRESENT | Block Cipher | SPN | 64 | 80/128 | 31 |
| 7-14 | PRINCE | Block Cipher | SPN | 64 | 128 | 12 |
| 8-15 | QARMA | Block Cipher | SPN | 64 | 64 | 27 |
| 9-16 | RECTANGLE | Block Cipher | SPN | 64 | 128 | 25 |
| 10-17…26 | SIMON | Block Cipher | Feistel | 32…128 | 64…256 | 32…72 |
| 11-27…36 | SPECK | Block Cipher | Addition/Rotation/XOR (ARX) | 32…128 | 64…256 | 22…34 |
| 12-37/38 | TWINE | Block Cipher | Type-2 Generalized Feistel Network (GFN-2) | 64 | 80/128 | 36 |
| 13-39 | XTEA | Block Cipher | ARX | 64 | 128 | 64 |

**Table 2.** NIST 2nd-Round Candidates, 32 families of 80 algorithms.

| Number | Algorithm | Kind | Block Size (bits) | Key Size (bits) | Num of Rounds | Algorithm Info |
|---|---|---|---|---|---|---|
| 1 | **ACE** | Authenticated encryption with associated data (AEAD)-block cipher-hash algorithm | 64 | 128 | 8 | Tag size of 128 bits, digest (hash) of 256 bits |
| 2 | **ASCON** | Authenticated encryption (AE)-block cipher and hashing | 64-128 | 128 | 30/32 | Key size = tag size = security level 128 bits |
| 3 | **COMET** | AE-block cipher | 64-128 | 128 | 27/80 | Block cipher mode of operation |
| 4 | **DryGASCON** | AEAD-block cipher-hash algorithm | 64-128 | 128/160/256 | 48 | N/A |
| 5 | **Elephant as (Dumbo-Jumbo-Delirium)** | AE Family-block cipher | Tweakable block cipher | 128 | 18/80/90 | Nonce-based encrypt-then-MAC construction |
| 6 | **ESTATE** | AE Family-block cipher | Tweakable | 128 | 40 | Block cipher-based MAC then encrypt |
| 7 | **ForkAE** | AE Family-block cipher | 64 | 128 | 53/75/87 | SKINNY primitive |
| 8 | **GIFT-COFB** | AE Family-block cipher | Tweakable | 128 | 40 | block cipher cryptographic primitive |
| 9 | **Gimli** | AE-block cipher and hashing | Tweakable | 256 | 24 | Cipher: 256-bit key, 128-bit nonce, 128-bit tag Hash: with 256-bit output |
| 10 | **Grain-128AEAD** | AEAD stream cipher | Tweakable | 128 | 160 | Nonce (IV) of size 96 bits and key of size 128 bits |
| 11 | **HyENA** | AE Family-block cipher | Tweakable | 128 | 40 | Hybrid feedback based |
| 12 | **ISAP** | AE Family-block cipher | 64 | 128 | 32/33/48/56 | Nonce-based AEAD |
| 13 | **KNOT** | AEAD-block cipher-hash algorithm | Tweakable | 128/192/256 | 118/136/160/208 | Permutation-based and bit-sliced AEAD and hashing algorithms |
| 14 | **LOTUS-AEAD and LOCUS-AEAD** | AEAD-block cipher | 64 | 128 | 28 | Block cipher-based AE scheme that employs OTR style |
| 15 | **mixFeed** | AE Family-block cipher | 64 | 128 | | Based on any block cipher with some key scheduling |
| 16 | **ORANGE** | Sponge AE-block cipher and sponge hash | 128 | 128 | 12 | Based on any permutation |
| 17 | **Oribatida** | AE Family-block cipher | Tweakable | 128 | 26/34 | Keyed permutation-based mode |
| 18 | **PHOTON-Beetle** | AE-block cipher and hashing | Tweakable | 128 | 12 | Sponge-based mode Beetle with the P256 (used for the PHOTON hash) |

**Table 2.** *Cont.*

| Number | Algorithm | Kind | Block Size (bits) | Key Size (bits) | Num of Rounds | Algorithm Info |
|--------|-----------|------|-------------------|-----------------|---------------|----------------|
| 19 | **Pyjamask** | Block cipher | 96/128 | 128 | 14 | SPN |
| 20 | **Romulus** | AE Family-block cipher | Tweakable | 128 | 48/56 | Based on a Tweakable block cipher (TBC) Skinny |
| 21 | **SAEAES** | AE Family-block cipher | 128/192/ 256 | 128 | 10 | AES-based AEAD |
| 22 | **Saturnin** | MAC-block cipher-hash | 256 | 256 | 10 | N/A |
| 23 | **SKINNY** | AEAD-block cipher-hash algorithm | Tweakable | 128 | 48/56 | Tweakable block ciphers |
| 24 | **SPARKLE (SCHWAEMM and ESCH)** | AE-block cipher and hashing | 128/192/ 256 | 128/196 /256 | 10/11/12 | Permutations based on an ARX design |
| 25 | **SPIX** | MAC-block cipher | 64 | 128 | 72/144 | Hybrid |
| 26 | **SpoC** | AEAD-block cipher | 192/256 | 128 | 108/ 144 | Permutation-based mode |
| 27 | **Spook** | AEAD-block cipher | Tweakable | 128/256 | | Sponge based |
| 28 | **Subterranean 2.0** | Hashing, MAC computation, stream encryption, AE | 128 | 128 | | N/A |
| 29 | **SUNDAE-GIFT** | Block cipher | 128 | 128 | 40 | N/A |
| 30 | **TinyJambu** | AE-block cipher | 32 | 128/192/256 | 8 | Based on a keyed permutation |
| 31 | **WAGE** | AEAD-stream cipher | 64 | 128 | 111 | Permutation based on the Welch–Gong (WG) stream cipher |
| 32 | **Xoodyak** | Hashing, encryption, MAC, AE-block cipher | 128 | 128 | 12 | Duplex object |

### 3.2. Compilation

This study was based on C language implementation as low language to reach an adequate elimination of any software barrier between algorithms' implementation and execution. Hence, the study uses MinGW [35] as a container of GNU compiler collection (GCC) [36]. MinGW is a free and open-source software development environment that includes GNU compiler collection (GCC) and its libraries. It is used in this study in the Linux operating system of the Raspberry Pi platform. As for Arduino platform, Arduino-Integrated Development Environment (IDE) [37] is the best integration, superseding the compiling and execution in this study.

### 3.3. Measuring Concepts and Metrics

Measurements for lightweight cryptography study were gathered depending on related works and other similar research studies that analyzed and compared lightweight cryptography ciphers. These measuring concepts and their metrics are summarized in Table 3.

**Table 3.** Measurement of LWC and their metrics.

| Measurement | Metrics (in) | Tool of Measuring for Arduino | Tool of Measuring for Raspberry Pi |
|---|---|---|---|
| Key size | bits | Algorithm specs | Algorithm specs |
| Block size | bits | Algorithm specs | Algorithm specs |
| Rounds number | number of rounds | Algorithm specs | Algorithm specs |
| ROM occupation | bits or bytes | Arduino IDE | Size command |
| RAM occupation | bits or bytes | Arduino IDE | Valgrind |
| Code size | Kbytes | Size occupied on memory | Size occupied on memory |
| Encryption (ENC) or decryption (DEC) speed throughput | Bytes/s | Programming + Equation (1) | Programming + Equation (1) |
| ENC or DEC speed latency | cycle/Block | Programming + Equation (3) | Programming + Equation (3) |
| Key schedule speed throughput | Bytes/s | Programming + Equation (2) | Programming + Equation (2) |
| Key schedule speed latency | cycle/Block | Programming + Equation (4) | Programming + Equation (4) |
| ENC or DEC power (throughput) | joules/s | Current (power) sensor | Current (power) sensor |
| ENC or DEC energy (latency) | joules/bit | Current (power) sensor + Equation (7) | Current (power) sensor + Equation (7) |

Below is the briefing and the equations used in Table 3:

1.  Arduino IDE is used during the uploading phase of C codes onto Arduino Board to measure:

    - ROM occupation: by observing "program storage space" where the Arduino sketch is stored.
    - RAM occupation: by observing the unused space for local variables, then the used space would indicate the "global variable" of dynamic memory that is the SRAM (static random-access memory), which is where the sketch creates and manipulates variables when it runs.

2.  Encryption and decryption speed throughput are measured in bytes/s in both platforms through programming loops and Equation (1).

$$\frac{Ps}{\tau}(\text{bytes/s}) \tag{1}$$

*Ps* is the size of text in ENC or DEC in bits.
$\tau$ is the time taken during one ENC or DEC.

3.  Key schedule speed throughput concerning key expansion and Equation (2).

$$\frac{Ks}{\tau}(\text{bytes/s}) \tag{2}$$

*Ks* is the size of text in expanded key in bits.

4.  ENC and DEC speed latency are measured in B/s
    (cycles/block) by using speed throughput Equation (2):

$$\frac{f}{(1)/Bs}(Cycles/Block) = \frac{f * Bs * \tau}{Ps}(Cycles/Block) \tag{3}$$

*f* is the processor frequency in hertz.
*Bs* is the block size of the algorithm in bytes.

5.  Key scheduling speed latency derived from Equation (2):

$$\frac{f * K * \tau}{Ks}(Cycles/Block) \tag{4}$$

*K* is key size of the algorithm in bytes.

6.  ENC and DEC power (throughput) measured in joules/s (j/s) by using a current sensor (power sensor).

7.  ENC and DEC energy (power **latency**) in joules/bit is measured by using energy throughput and speed throughput as:

$$\frac{ETh(j/s)}{(1) * 8}(Jouls/bit) = \frac{ETh * \tau}{Ps * 8}(Jouls/bit) \tag{5}$$

*ETh* is energy throughput in j/s of ENC and DEC.

### 3.4. Methodology

1.  The following methods were used while benchmarking the different metrics for the selected cryptographic algorithms:

    - Different Cryptographic Algorithms, Same Platform: Comparing different algorithms on the same platform is performed by measuring the throughput in bytes per second that would be satisfactory.
    - Same Cryptographic Algorithms, Different Platform: Looking at timing information mainly has one of two incitements, either the interest in comparing the performances of two algorithms or the quantity of information being processed through a particular platform. However, measuring in bytes per second has no actual indication of algorithm performance on different platforms. Hence, it is preferred to measure the processor clock cycle during the processing task of each byte, indicated as cycles per bytes, which allows for more relevant comparisons. It is performed simply by dividing the clock speed in hertz which is the cycles per second (C/S) by the speed throughput of the algorithm in bytes per second (B/S), yielding cycles per byte (C/B).
    - Different Cryptographic Algorithms, Different Platform: The difference in key size and block among cryptographic algorithms can be assessed using bytes per second as a metric of measurement in the same platform. However, this cannot

be used in different platforms (and neither can cycles per byte). Here, comes the notion of using cycle per block as a comprehensive measure of comparison between them.

2. Relative Reference (RR): Nevertheless, the last comparison measurement is restrained by the enormous difference in results, speed throughputs, speed latency, and even that of energy, as will be seen in Section 4. The relative reference algorithm (RR) is used by taking the percentage of each measurement result (latency and throughput) of the algorithms, then comparing them to 1 or 100%. On that account, AES-128block-128key, AES-128-192, and AES-128-256 are selected, the traditional standard, which will be a notable solution in this kind of comparison.

3. Repeating the Experiments: For numerous distinctive reasons, it can be decently troublesome to obtain measuring results such as time and speed accurately in a single iteration of coding. Frequently, internal clocks that the software or executed program can read have some degree of asynchronous precision from the core processor clock. More essentially, there is regularly a critical overhead included in such measuring results, such as the cost of context switches and sometimes timing overhead. That is, finding measurement of algorithms in this context should avoid procedure call overhead. One method is to run the algorithm many times like loops in coding, then averaging the total time to acquire the best indication of overall performance results. Furthermore, the repetition of encryption or decryption process would smooth out random effects such as IRQ (Interrupt request) signal due to external activity by adjusting the loop to an experimental number attained.

On that account, the formulas of speed throughput of ENC and DEC followed by key schedule formulas would be refined as:

$$\frac{Ps * Nl}{\tau}\text{(bytes/s)} \tag{6}$$

$$\frac{Ks * Nl}{\tau}\text{(bytes/s)} \tag{7}$$

*Nl* is number of loops.

4. Mean and Standard Deviation: As mentioned before, the algorithms' speed and power are the averages (averaging the total time) of many running times in loops (1000 times). This process is repeated 100 to 1000 times as needed to reach an acceptable standard deviation of the averages obtained (1000 of 1000). The means and standard deviations related to speed and power, including key schedule speed of the algorithms, are provided in the github.

5. Programming Libraries: We have implemented the needed formulas in one programming library named metrics.h for both platforms. The metrics.h library contains the implementation of Equations (3)–(7) for exclusive grouping results of the algorithms in one hand pack. In addition, changing the number of loops would be very easy using such a method for simplicity in work and other tasks. Furthermore, for the Raspberry Pi platform, the metrics.hlibrary includes the exporting of the result in ".csv" (a mode of extension file) in one hit for all studied algorithms in speed measures that would help significantly in any re-benchmarking of the algorithms when needed. Meanwhile, ".csv" exporting cannot work for the Arduino platform in programming. A software tool as an add-on for Microsoft Excel is used called PLX-DAQ [38]. Parallax Data Acquisition tool (PLX-DAQ) is a software that drops the numbers into columns as they arrive from the PC's serial port and it has the following features:

- Plot or graph data as it arrives in real-time using Microsoft Excel.
- Record up to 26 columns of data.
- Mark data with real time (hh:mm:ss) or seconds since reset.
- Read/write any cell on a worksheet.
- Baud rates up to 128K.

Besides that, PLX-DAQ has shown a significant benefit in measuring power and graphical observation of it.

6. Power Connection: Power (energy) is measured through the use of the Adafruit INA219 [39] current sensor. This measuring process requires another Arduino board to read the sensor data of measuring that is fed from the load to the platforms (Arduino UNO and Raspberry Pi). The second Arduino board is used as a current-voltage monitor. The monitoring process is combined by PLX-DAQ software tool through EXCEL with the necessary Arduino programming. The circuit connection is shown in Figure 1.

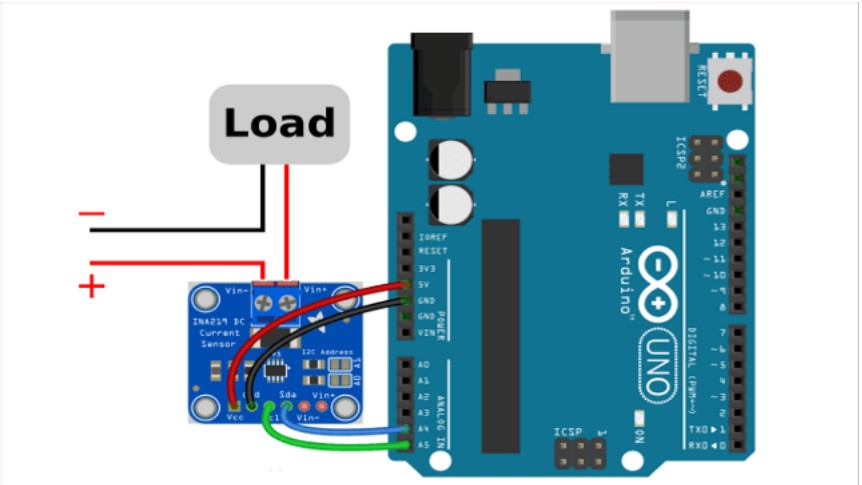

**Figure 1.** INA219 wiring with the second Arduino board.

7. Hardware Platform: We used two different hardware platforms in this study for comparison and benchmarking purposes:

   - ATMEGA328P is a single-chip microcontroller of the megaAVR family with an 8-bit RISC processor core architecture. It is used in basic Arduino boards, such as Arduino UNO.
   - Raspberry Pi 3 Model B V1.2 is the third generation of Raspberry Pi.
   - Ammeter (power measurement) used: Power (energy) is measured through the use of the Adafruit INA219 current sensor. One advantage about this sensor is that it is inserted on the "high side" of the circuit instead of the ground side, which is how many of those voltage and current display modules are wired. That makes it ideal for use as a voltage and current display. It operates on a power supply of 5 volts, which can be supplied by the Arduino UNO too.

8. Methodology for conducting the performance analysis The methodology for conducting a performance analysis of lightweight cryptographic algorithms implemented on a Raspberry Pi or Arduino UNO can involve the following steps:

   (a) Selection of algorithms: Identify and select a set of lightweight cryptographic algorithms that are relevant to the application and that can be implemented on a Raspberry Pi/UNO.

   (b) Implementation: Implement the selected algorithms on a Raspberry Pi/UNO using a programming language such as Python or C. Data collection: Collect performance data by running the implemented algorithms on the Raspberry Pi and measuring the execution time and memory usage for each algorithm.

   (c) Data analysis: Analyze the collected data to determine which algorithms are the most efficient and secure for the given application. This may involve comparing the execution time and memory usage of the algorithms, as well as analyzing any security weaknesses or vulnerabilities in the algorithms.

(d)  Optimization: Based on the analysis, optimize the implementation of the chosen algorithm to minimize resource usage and increase security. Reporting: Prepare a report detailing the performance analysis and optimization results, including any recommendations for further research or improvement. The whole process is illustrated below:

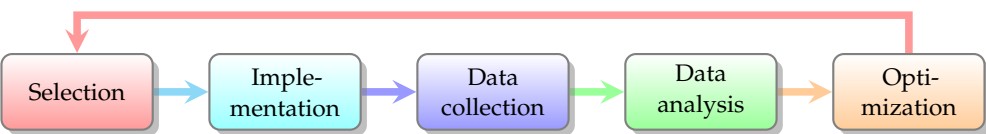

## 4. Results

This section presents the main benchmarking observations. For each metric, the best ten and the worst five performing algorithms are selected in a quick and brief overview. The focus will be on the following measurements:

*   The ROM, RAM, code size, and key schedule speed throughput and latency of the 39 ciphers.
*   The number of rounds, encryption/decryption speed throughput/latency, and energy throughput/latency of all the 119 ciphers.

In addition, the measurement tools used are stated briefly for each metric.

1.  Analysis of the number of rounds: As part of the algorithm design, modern ciphers increase their security (confusion and diffusion) through the repeated execution (n times) of a simple round function. In block ciphers, the input and output of the round function are equal to the cipher block size in general. As a standard rule, increasing the number of rounds *n* increases the security level, while decreasing the number of rounds would play a significant role in shortening the execution time of the encryption and decryption, which is one of the essences of lightweight cryptography. In this project, the range interval of the number of rounds of the studied algorithms is [8, 254], as shown in Figure 2. Among the five ciphers with the largest number of rounds, the Katan–Ktantan family is designed with the highest, while the Ace-64-128 and TinyJambu families are with the least among the smallest 10.

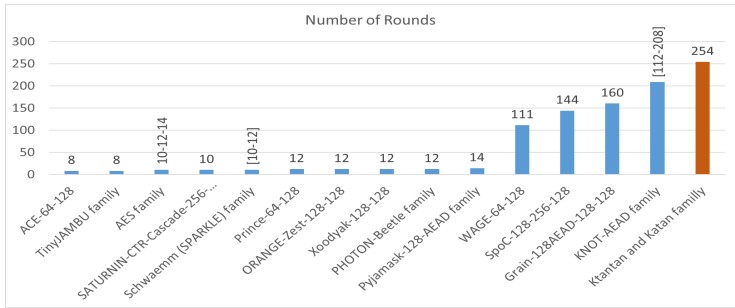

**Figure 2.** 19 algorithms of smallest N of rounds and 13 of the biggest.

2.  Analysis of code size: The code size is the size of the algorithm code written in C language. It is the occupied space on the disk or memory. Moreover, it might project the occupation of the ROM and RAM sizes. Many of the cipher implementations were optimized in coding for different considerations, among them, of significant importance, was changing the plaintext declaration in a way that takes no more space than what is required. According to the results illustrated in Figure 3, the range intervals of the code sizes are [3.6, 21] and [3.49, 17.3] in Kbytes for UNO and Pi, respectively. Of the 10 smallest code sizes, the least are Katan-32-80 and Present-64-128, whereas the biggest are Piccolo-64-80 and Rectangle-64-128 in UNO and Pi, respectively.

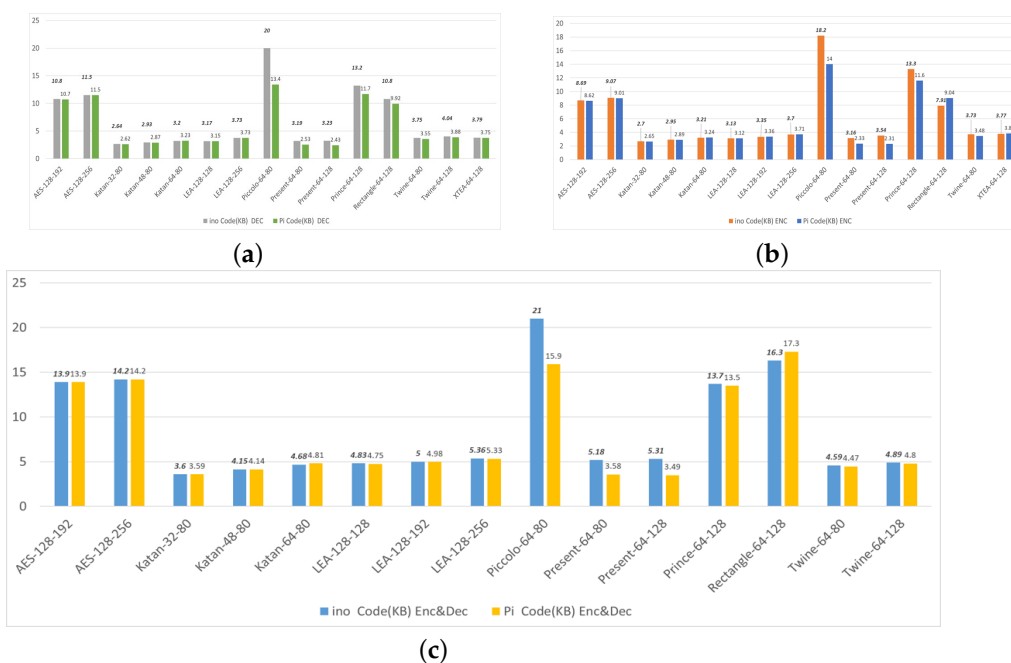

**Figure 3.** Code size of algorithms used in benchmarking. (**a**) The 10 algorithms of DEC smallest code sizes and 5 of largest. (**b**) The 10 algorithms of smallest ENC code sizes and 5 of largest. (**c**) The 10 Algorithms of smallest code sizes and 5 of largest.

3.  Analysis of ROM size: Read-only memory or ROM is a non-volatile memory. Data stored in ROM are generally the code and the tables that do not need modifications. The Arduino Uno (ATMEGA328P) ROM size (Flash) used is 32 Kbytes in size, while the Pi's is a variable SD card where 64 Gbytes is used in this project. The Arduino IDE fulfills the ROM measurement requirement that gives the occupied size of the storage space. However, in Pi, the "size" command is used in the terminal (Linux OS) as a tool to obtain the text, data, Block Started by Symbol (BSS) (Block Started by Symbol is the space that contains all the uninitialized data), and DEC sizes (DEC = text + data + BSS) of the compiled code file in an Extensible Linking Format (ELF) extension. The range interval is [1.48, 6.05] and [3.81, 9.67] in Kbytes for UNO and Pi, respectively. Figures 4 and 5 show the least cipher-demanding ROM among the 10 smallest which are the LEA family/Prince-64-128 and Present-64-80, respectively, for UNO and Pi, while the biggest demand is Piccolo-64-80 for both platforms.

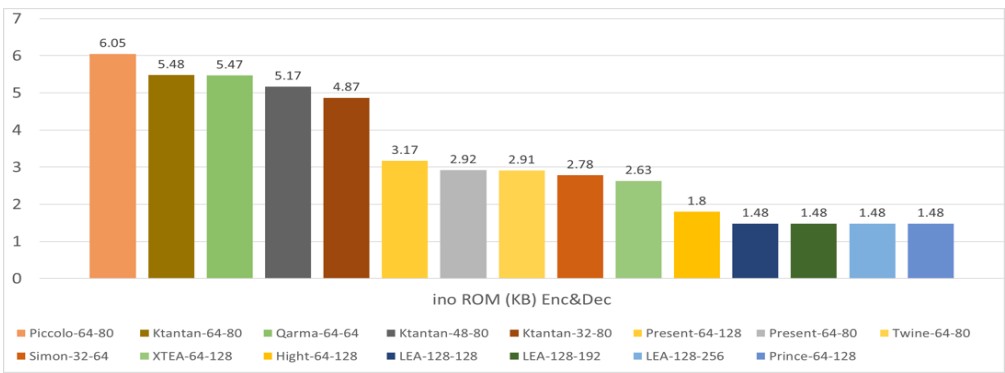

**Figure 4.** The 10 smallest ciphers demanding ROM and 5 largest for UNO.

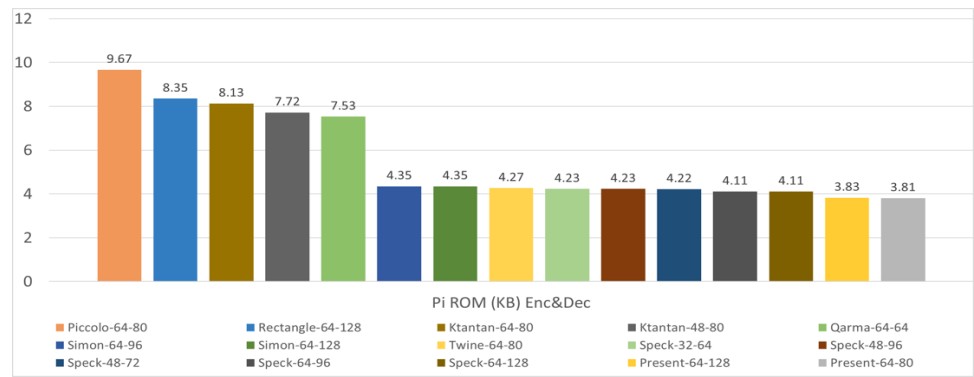

**Figure 5.** The 10 smallest ciphers demanding ROM and 5 largest for PI.

4.  Analysis of RAM occupation: Random-access memory (RAM) is the short-term memory where data are stored to be processed by the processor. Data are stored in the RAM in the form of a heap and stack. The size of UNO's RAM (SRAM) used is 2Kbytes, whereas the Pi's is 1Gbytes. The Arduino IDE is sufficient for obtaining the RAM usage by the ciphers. However, the Valgrind [40] tool is used to measure the RAM occupation in Pi. Valgrind is a tool suite for debugging and profiling, and the Massif profiler tool of the Valgrind is used to measure the RAM. Figures 6 and 7 show the 10 smallest and 5 largest RAM size occupations in UNO and Pi, respectively. As it can be seen, the range intervals are [264, 994] in bytes and [1.03, 21.34] in Kbytes of UNO and Pi, respectively. The least are the LEA and the Simon–Speck families, and the largest are the Ktantan family and Rectangle-64-128 in UNO and Pi, respectively.

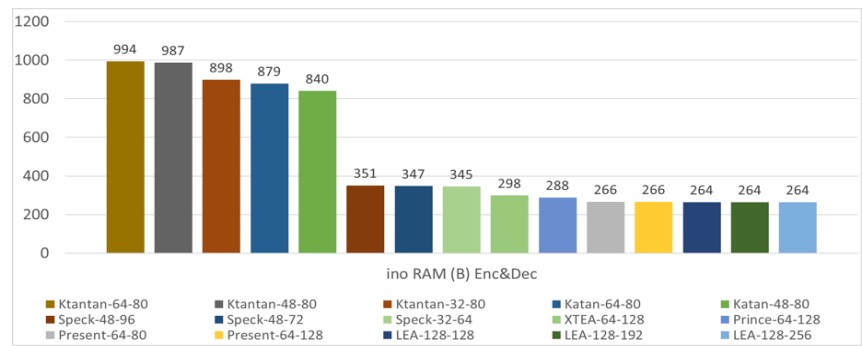

**Figure 6.** The 10 smallest ciphers RAM usage and 5 largest for UNO.

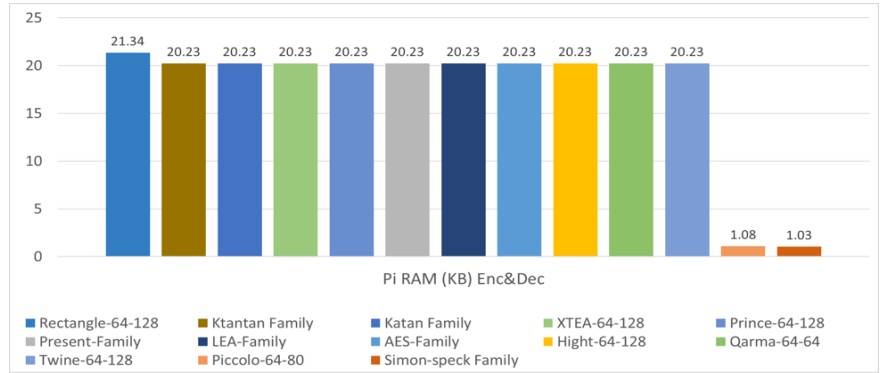

**Figure 7.** The 10 smallest ciphers RAM usage and 5 largest for PI.

5.  Key scheduling speed: The key schedule is a process of key expansion that expands a short key (40 to 256 bits) to a larger key (or to a number of round keys) (100 to 1000 s bits) for use in encryption and decryption algorithms. This process in ciphers has a direct impact on the security of the cipher. The key schedule speed was

measured using C programming routines that calculate the time taken. Formulas (2) and (4) mentioned in Section 3.3 are used for the throughput in bytes per second and latency in cycles per block, respectively. Figures 8–11 exhibit the 10 largest and 5 smallest algorithms for the key schedule speed in UNO and Pi, respectively. The range intervals are [0.99, 400,000] in Kbytes/s and [0.029, 106.667] in Gbytes/s of UNO and Pi, respectively, for the throughput. The best is Prince-64-128, and the worst is the Ktantan family in both UNO and Pi. Further, the range intervals for the latency are [0.00032, 161.34] in Kcycles/block and [0.18, 412.02] in cycles/block of UNO and Pi, respectively. The best is XTEA-64-128, and the worst is the Ktantan family in both UNO and Pi.

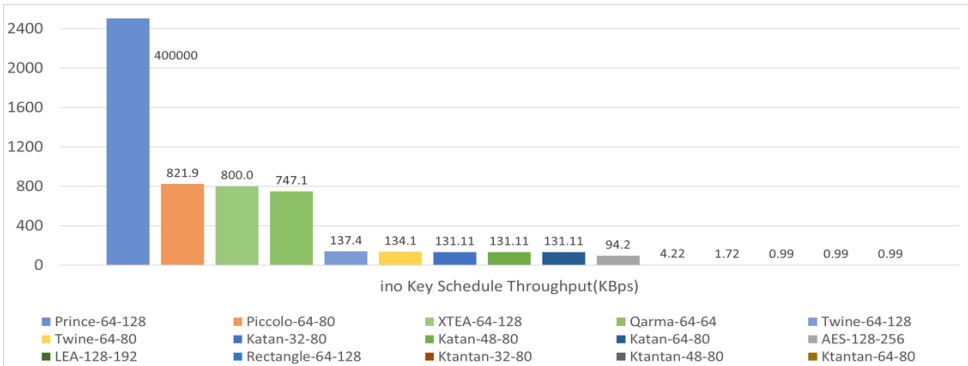

**Figure 8.** The 10 largest speed throughput in key schedule and 5 least in UNO.

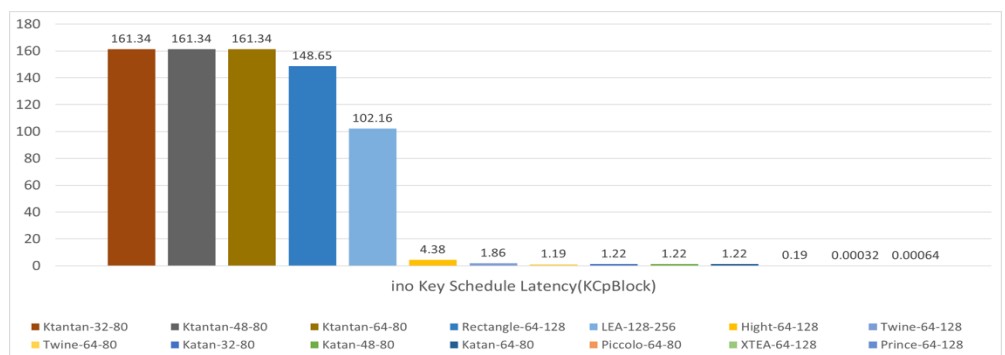

**Figure 9.** The 10 least speed latency in key schedule and 5 biggest in UNO.

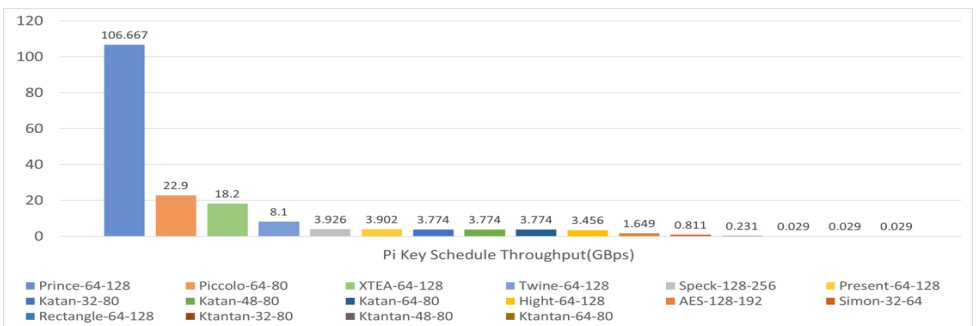

**Figure 10.** The 10 largest speed throughput in key schedule and 5 least in Pi.

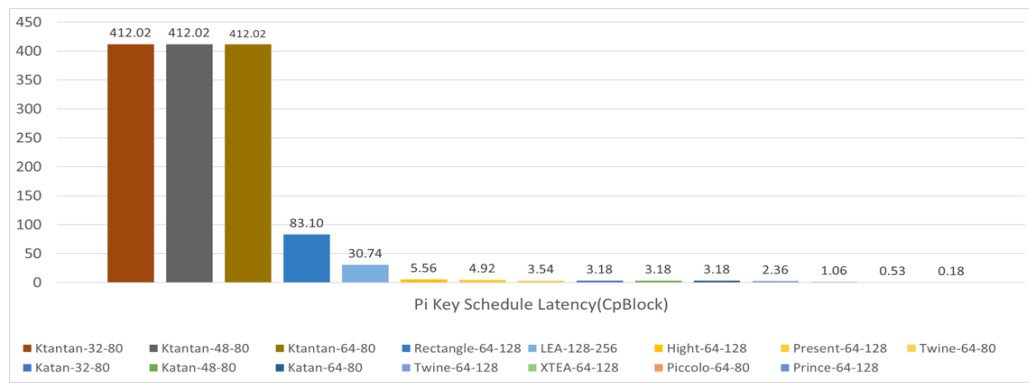

**Figure 11.** The 10 least speed latency in key schedule and 5 biggest in Pi.

6.  Encryption and decryption speed: The encryption or decryption speed measuring tool is by the internal programming (C) of the time taken. Formulas (6) and (3) mentioned in Section 3.3 are used for the throughput in bytes per second and latency in cycles per block, respectively.

    Figures 12–15 reveal the 10 largest and 5 smallest algorithms for the ENC speed in UNO and Pi, respectively. The range intervals are [0.1, 64.1] in Kbytes/s and [0.009, 6.99] in Gbytes/s of UNO and Pi, respectively, for the throughput. The best are Hight-64-128 and LEA-128-128; however, the worst are Jumbo-128-128 and Ktantan-32-80 in UNO and Pi, respectively (here the biggest is the fastest). The range intervals for the latency are [2, 2490.24] in Kcycles/block and [1.44, 864.65] in cycles/block of UNO and Pi, respectively (for the latency, the smallest is the best and the fastest). The best are Hight-64-128 and Speck-48-72, though the worst are Jumbo-128-128 and ISAP-K-128-64-128 in UNO and Pi, respectively.

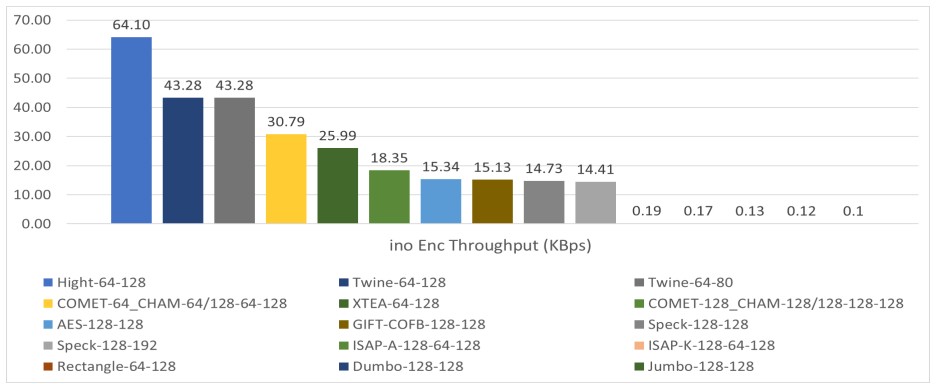

**Figure 12.** The 10 largest speed throughput in ENC and 5 least in UNO.

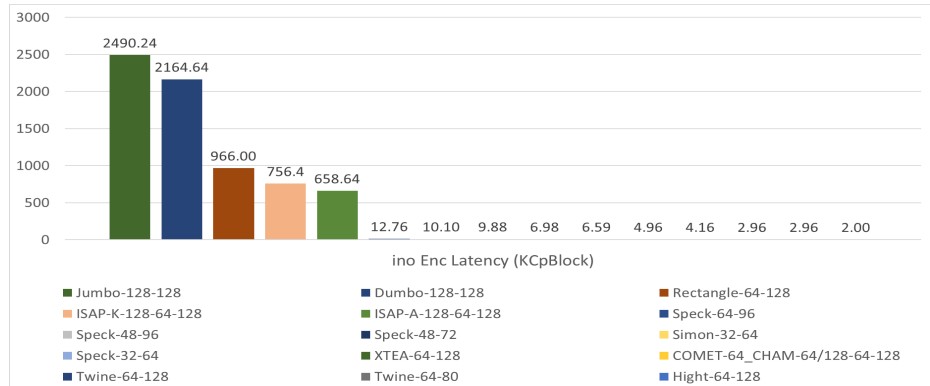

**Figure 13.** The 10 least speed latency in ENC and 5 biggest in UNO.

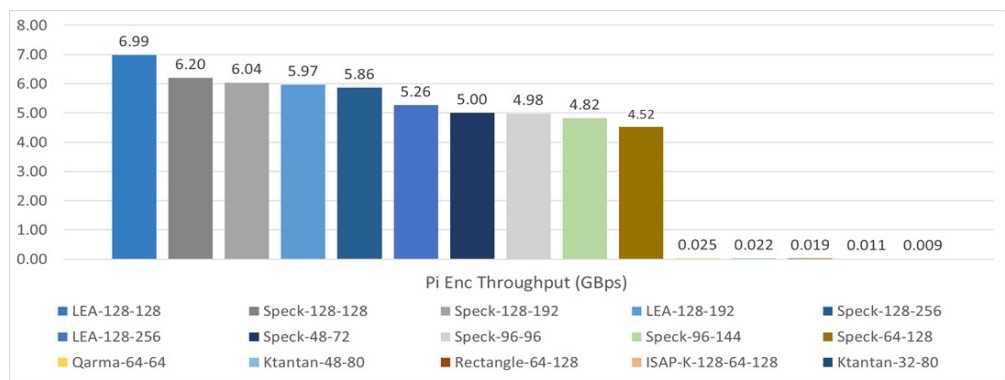

**Figure 14.** The 10 largest speed throughput in ENC and 5 least in Pi.

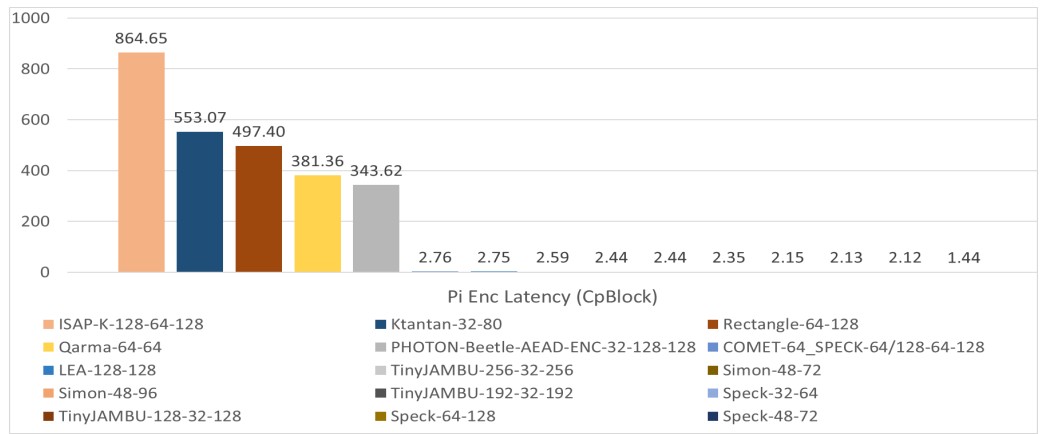

**Figure 15.** The 10 least speed latency in ENC and 5 biggest in Pi.

7. Encryption and decryption power and energy consumption: Energy utilization is the measure of the electrical effort used during the execution of an operation (algorithm), and the whole energy consumed is the time integral of the power. In lightweight cryptography, energy consumption per unit operation of the algorithm defines good metrics for designing. The measuring tool used for energy in this project is the Adafruit INA219 current sensor through a connection of a second Arduino board used as a current-voltage monitor. The monitoring process is combined by the PLX-DAQ software tool through EXCEL with the necessary Arduino programming. The provided Formula (5) in Section 3.3 is used for the energy in joules per byte while the mean power obtained from sensor monitoring is the power throughput in joules per second.

Figures 16–19 show the 10 smallest and 5 largest algorithms for the ENC power. The range intervals are [0.959, 14.28] in mj/s and [168, 303] in mj/s for the power. The least are SUNDAE-GIFT-0-128-128 and XTEA-64-128. However, the worst are Simon-128-256 and DryGASCON128k56-128-128 in UNO and Pi, respectively. The range intervals for the energy are [0.097, 67.4] in μj/byte and [35.4, 26,870] in nanoj/byte (or [0.035, 26.87] in μj/byte) of UNO and Pi, respectively. The least are SUNDAE-GIFT-0-128-128 and Speck-128-128, though the worst are Rectangle-64-128 and Ktantan-32-80 in UNO and Pi, respectively.

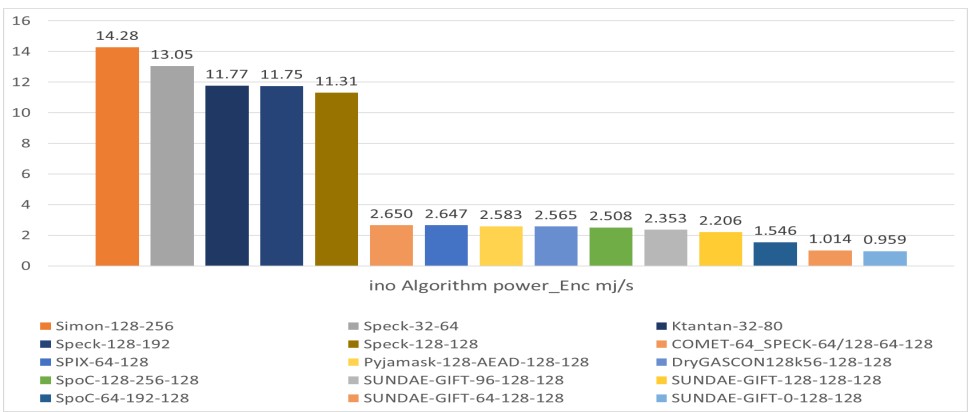

**Figure 16.** The 10 least power in ENC and 5 biggest in UNO.

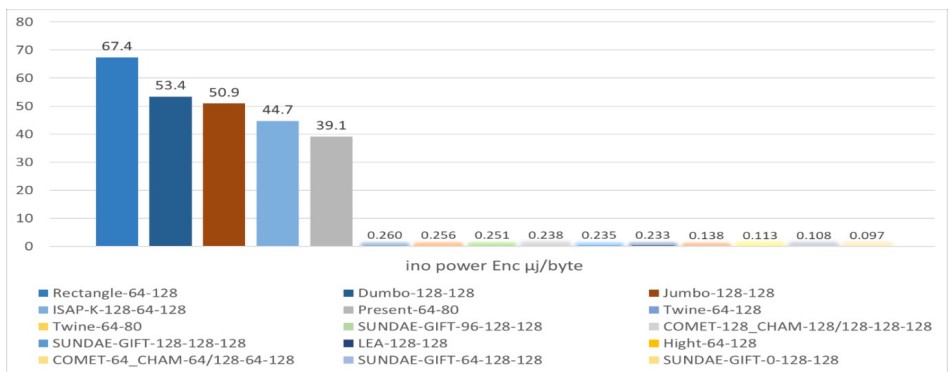

**Figure 17.** The 10 least energy in ENC and 5 biggest in UNO.

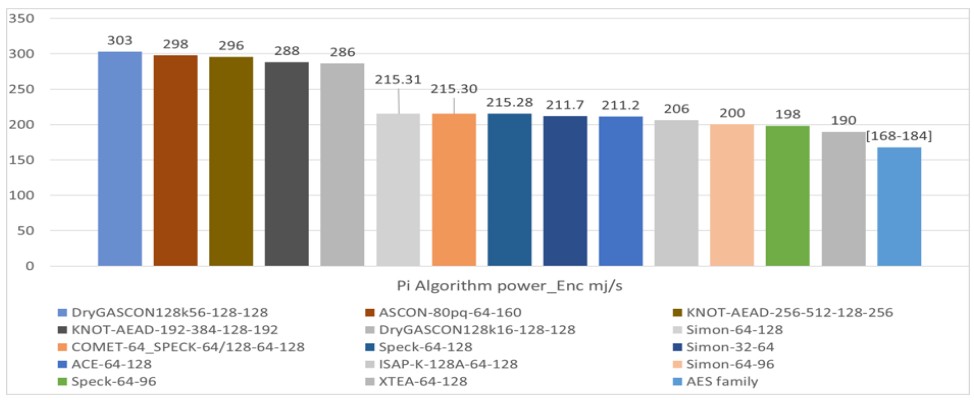

**Figure 18.** The 10 least power in ENC and 5 biggest in Pi.

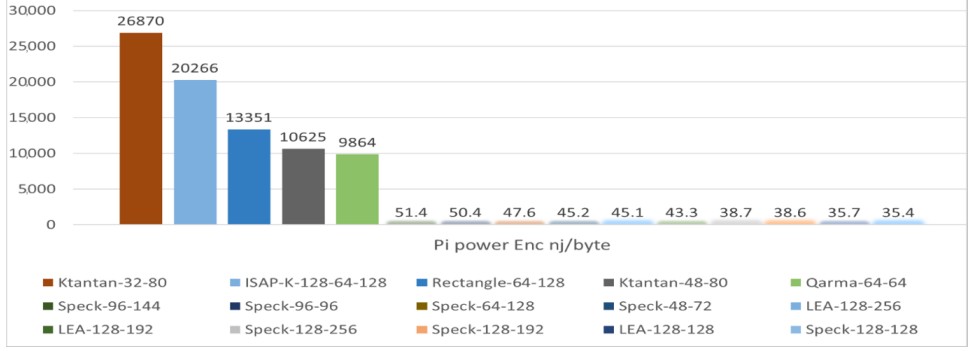

**Figure 19.** The 10 least energy in ENC and 5 biggest in Pi.

To summarize, we presented the benchmarking of the code size, ROM, RAM occupation, and key schedule for the first-part algorithms of 39. Then, we presented the benchmarking of the speed throughput, speed latency, power, and energy for the first combined by the second part of the algorithms to be 122 (plus the AES). The code size and ROM occupation are approximately the same in value with a slight change from the unit (Kbyte) point of view for both platforms, whereas the RAM, key schedule speed, ENC/DEC speed, and energy differ thousands of points (Kilo) with Pi taking the lead. In general, Raspberry Pi is scoring as the best in all measuring metrics throughout all the ciphers.

## 5. Discussion

Some of the algorithms' measured speed and power latencies (plus the key schedule) of the ENC are presented in the following figures as a percentage to the RR (AES). They are grouped in two ways: block or key sizes. Some groups with their corresponding figures are as follows:

- The 128-bits block size and 96-bits key size RR% groups of the key schedule speed latency presented in Figures 20 and 21.

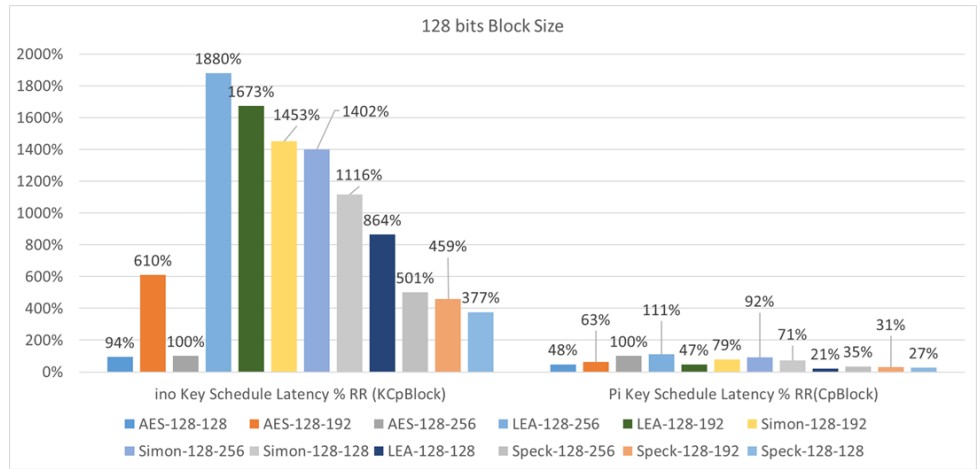

**Figure 20.** Group of 128-bits block size algorithms measuring key schedule speed latency with respect to RR% in UNO versus Pi.

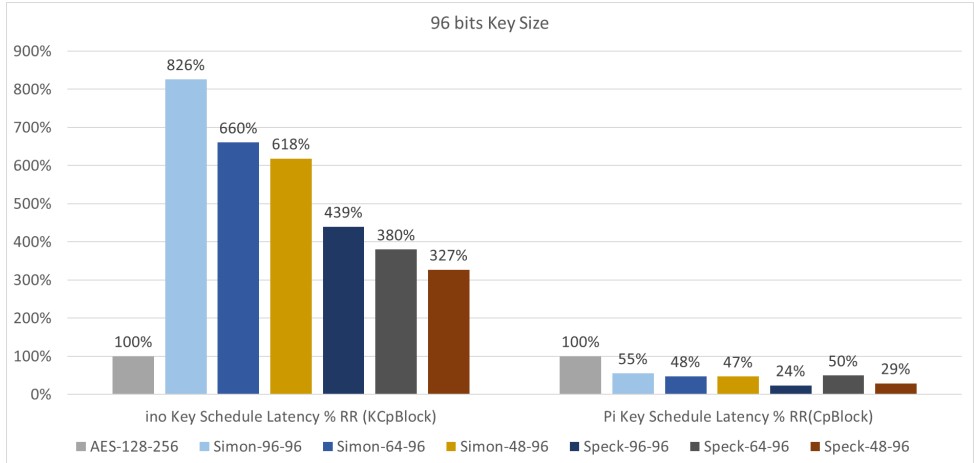

**Figure 21.** Group of 96-bits key size algorithms measuring key schedule speed latency with respect to RR% in UNO versus Pi.

- The 32-bits block size and 256-bits key size RR% groups of the ENC speed latency presented in Figures 22 and 23.

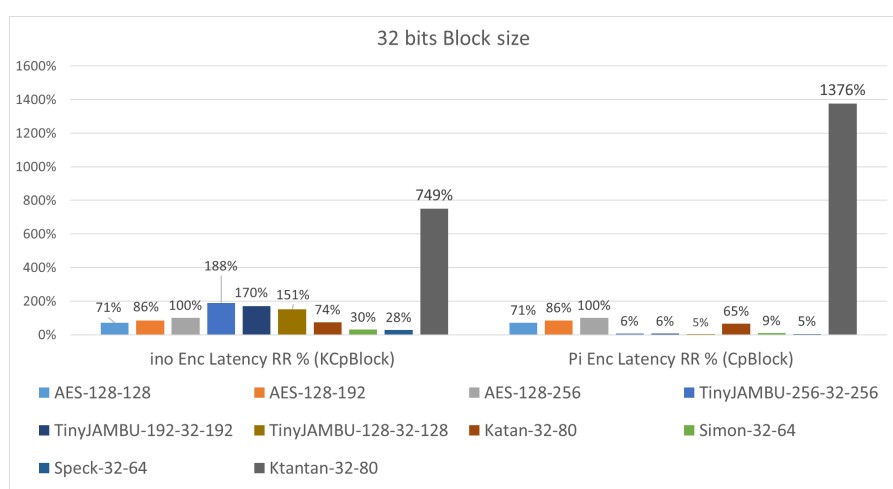

**Figure 22.** Group of 32-bits block size algorithms measuring ENC speed latency with respec to RR% in UNO versus Pi.

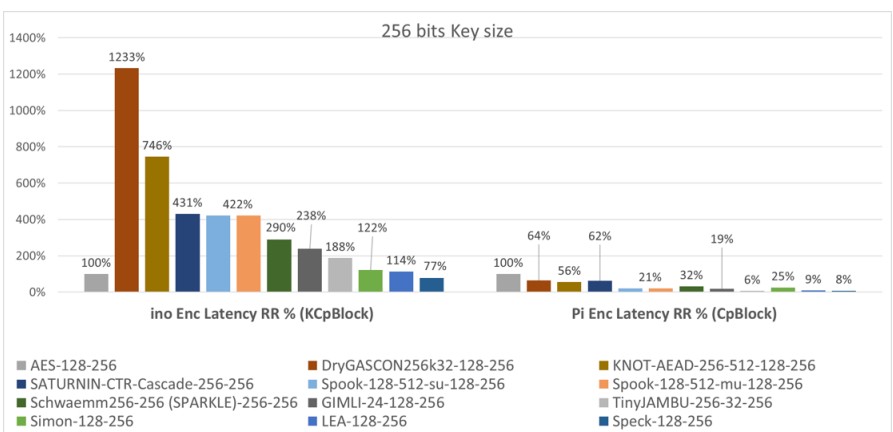

**Figure 23.** Group of 256-bits key size algorithms measuring ENC speed latency with respect to RR% in UNO versus Pi.

- The 128-bits block size (all of the 128-bits key size) and 80-bits key size RR% groups of ENC energy (power latency) presented in Figures 24 and 25.

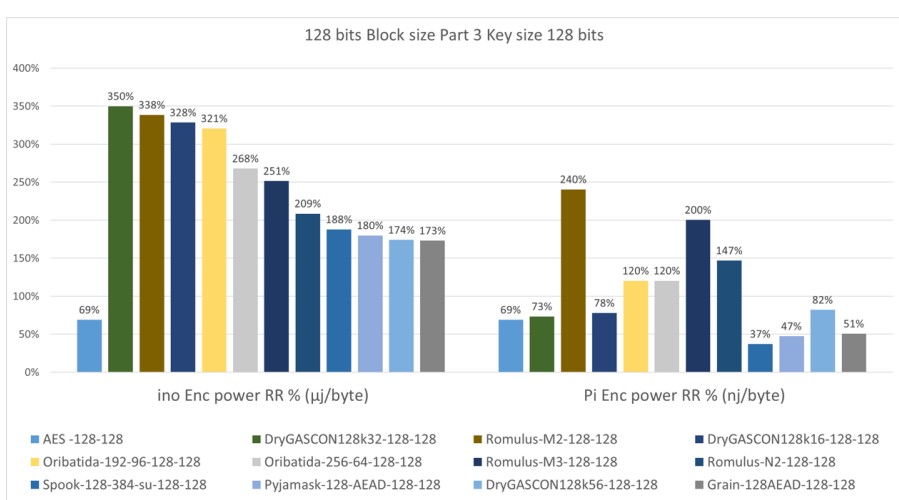

**Figure 24.** Group of 128-bits block and key sizes algorithms measuring ENC energy with respect to RR% in UNO versus Pi.

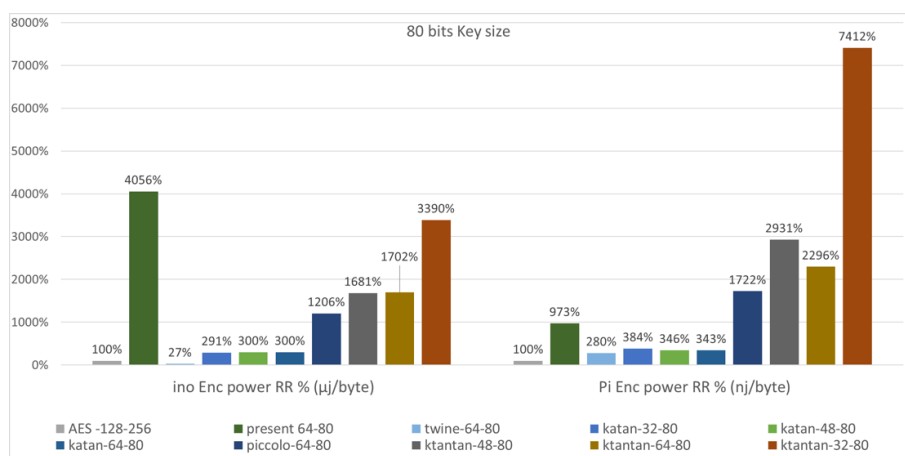

**Figure 25.** Group of 80-bits key size algorithms measuring ENC energy with respect to RR% in UNO versus Pi.

After an adequate observation over all these groupings and their results between UNO and Pi, it was realized that for more than 85% of the algorithm throughout, the metrics used had gone better in Pi. For example, Lea-128-192 was 1673% to the RR in the key schedule for UNO, whereas it became 47% for Pi. Statistically, in the key schedule, there were 10 algorithms below 100% for UNO, while they became 33 algorithms for Pi; in the ENC speed latency, there were 28 algorithms below 100% for UNO, whereas the number became 79 algorithms for Pi; and in the ENC energy, there were 33 algorithms below 100% for UNO, whereas the number became 65 algorithms for Pi. Consequently, Raspberry Pi is revealing more lightweight features and behavior of the designed lightweight algorithms that might be explained "because of the Hardware and Software Architecture of the Raspberry Pi". The 67 winners in each group of the block or key arrangement throughout the key scheduling speed latency, ENC/DEC speed latency, and ENC/DEC energy are listed in Table 4. It can be concluded that 36 (84%) of 39 of the algorithms are faster (fewer cycles) in the key schedule, 106 (94%) of 110 of the algorithms are faster (fewer cycles) in the ENC, and 95 (86%) of 110 have taken less power in Pi compared to UNO, as shown in Table 4 with an additional statistical comparison.

**Table 4.** Overall comparison of algorithms' numbers and percentage in UNO versus Pi.

| Condition | Key Schedule Speed UNO (39) | Key Schedule Speed Pi (39) | ENC Speed UNO (110) | ENC Speed Pi (110) | Energy UNO (110) | Energy Pi (110) |
|---|---|---|---|---|---|---|
| Unit Normal Comparison | Kbytes/Kcycles per s | Gbytes/cycles per s | Kbytes/Kcycles per s | Gbytes/cycles per s | [0.097,67.4] µj/byte | [0.035,26.87] µj/byte |
| Overall sum of % | 30,760 | 6032 | 77,014 | 6114 | 68,138 | 42,308 |
| Overall sum of % decreases from UNO to Pi | ↘80% | | ↘92% | | ↘40% | |
| # of algorithms decreases in RR % | 3 (all of 64 Block size) | 36 (92%) | 6 (4 of key size 80) | 106 (94%) | 15 (8 of 9 of key size 80) | 95 (86%) |
| # of algorithms <100% in RR % | 10 (25%) | 33 (84%) (+23 added) | 28 (25%) | 79 (72%) (+51 added) | 33 (30%) | 65 (60%) (+32 added) |
| Overall sum of % | Is the sum of all the algorithms percentage in RR | | | | | |
| ↘ | The number of algorithms decreases | | | | | |
| <100% | Number of algorithms that are below 100% in RR | | | | | |

Moreover, it was realized as a quick observation that most of the algorithms that were worse in Pi compared against UNO were of an 80-bits key, like in the energy ENC. That may indicate a reliance on the 80-bits key in the algorithms' design is not preferable, but this should take further investigation to deduce such a conclusion.

A Score Table is presented in this section for some measuring metrics. The scores (or cards) are given to each algorithm in each of the selected measures: ROM, ENC speed

(throughput and latency), and ENC energy (throughput and latency). The algorithm with the least sum of all the scores would be considered as the best. For the first list of algorithms presented in Table 1, Figure 26a,b clearly show that LEA-128-128, OMET-64-CHAM-64-128, and Hight-64-128 are the best in UNO, while Speck-48-72, Speck-64-128, and XTEA-64-128 are the best in Pi. Regarding the final list presented in Table 2, Figure 26c,d clearly show that Schwaemm-256-128, GIFT-COFB-128-128, and Schwaemm-128-128 are the best in UNO, while Xoodyak-128-128, TinyJAMBU-192-32-192, and TinyJAMBU-128-32-128 are the best in Pi.

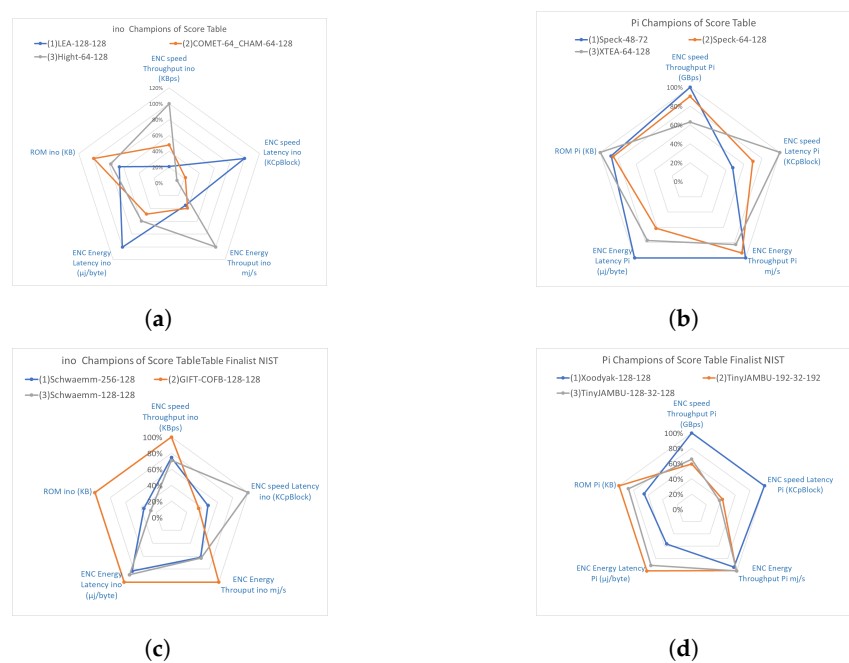

(a)  (b)

(c)  (d)

**Figure 26.** NIST best algorithms in UNO vs. Pi. (**a**) Best 3 For UNO. (**b**) Best 3 For Pi. (**c**) Best 3 for UNO. (**d**) Best 3 for Pi.

*Finalists of NIST*

NIST, in 2021, announced ten finalists as ASCON, Elephant, GIFT-COFB, Grain128-AEAD, ISAP, Photon-Beetle, Romulus, Sparkle, TinyJambu, and Xoodyak during the work of this project. These 10 families are of 28 algorithms of different block and key sizes that were presented before in Table 2. The corresponding % RR comparison between UNO and Pi of the best eight are presented in Figures 27 and 28 for the speed latency and energy. The best for the ENC latency are GIFT-COFB-128-128 and TinyJAMBU-128-32-128 in UNO and Pi, respectively. The best for the ENC energy is Schwaemm-128-128 in both UNO and Pi.

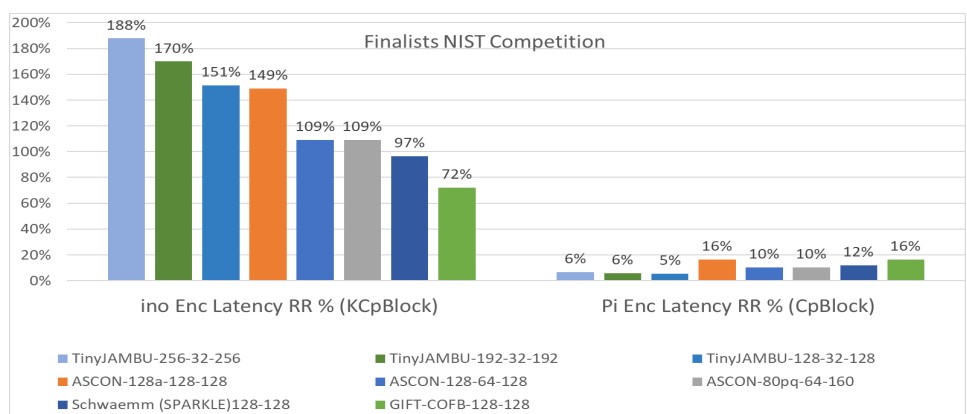

**Figure 27.** Best 8 Algorithms of NIST finalists in ENC speed latency with respect to RR % for both UNO and PI.

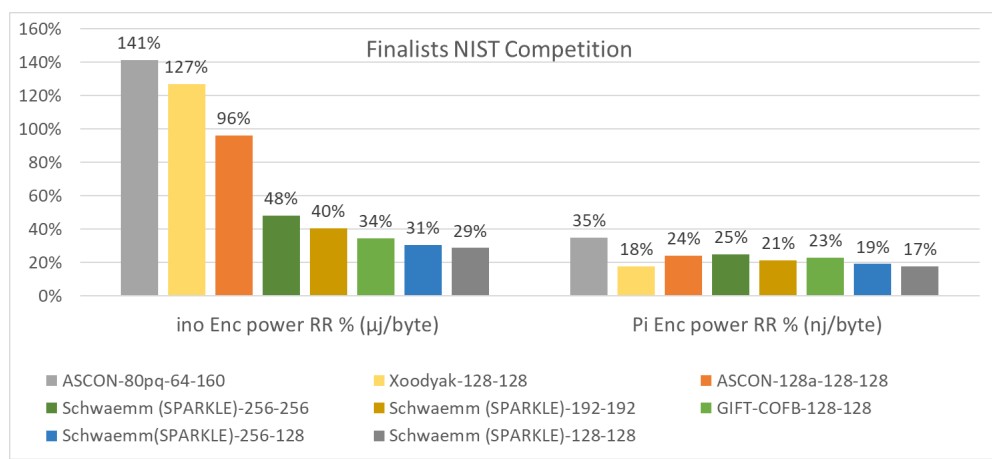

**Figure 28.** Best 8 Algorithms of NIST finalists in ENC energy with respect to RR % for both UNO and Pi.

In addition, the best three are presented using the "Score Table" in both UNO and Pi and are shown in Figures 29 and 30 as a radar graph. The best are Schwaemm-256-128, GIFT-COFB-128-128, and Schwaemm-128-128 in UNO and Xoodyak-128-128, TinyJAMBU-192-32-192, and TinyJAMBU-128-32-128 in Pi.

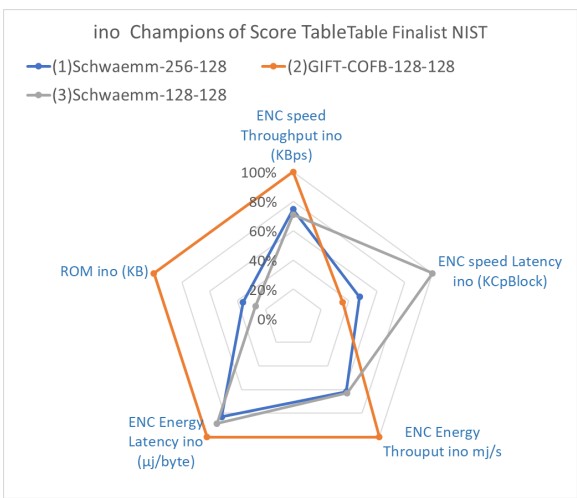

**Figure 29.** Radar graph of the three NIST finalists winners in 5 measures in UNO.

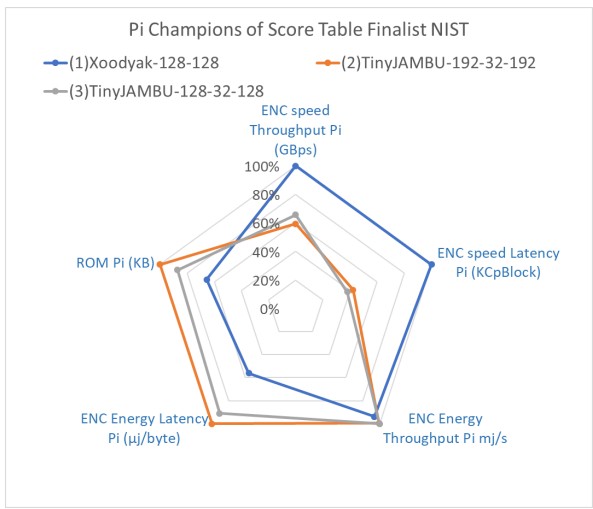

**Figure 30.** Radar graph of the three NIST finalists winners in 5 measures in Pi.

This section provided the analysis and results of 122 ciphers in the RR and Score Table plus the announced NIST finalists analysis alone. The RR percentage showed significant importance in a comparative analysis between the different platforms besides the latencies' metrics cycle per byte and joules per byte. Moreover, one of the reasons that worsens the algorithm is the high number of rounds, as presented in Figures 31 and 32, that provide the worst and best algorithms with their related number of rounds. During the RR analysis, it was realized that most of the algorithms that were worse in Pi than in UNO were of an 80-bits key size.

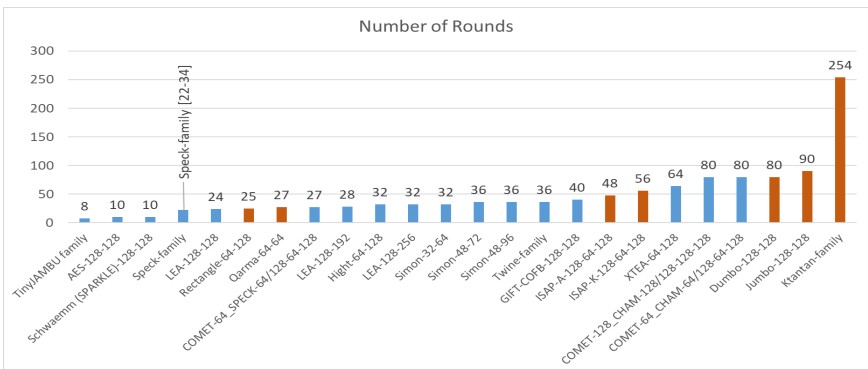

**Figure 31.** The best algorithms and worst throughout with their corresponding number of rounds. Brown color indicates the worst algorithms in previous measures.

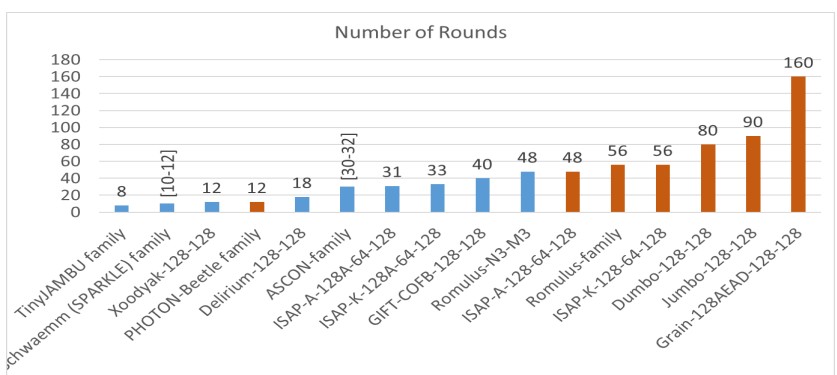

**Figure 32.** All algorithms of NIST finalists with their corresponding number of rounds. Brown color indicates the worst algorithms in previous measures.

During this work, a number of problems were encountered:

1.  The first problem encountered was the exceeding of the Arduino UNO ROM size by a handful of the algorithms. This issue led to the code optimizing of these algorithms. Some simple code optimization and changes in the declaration minimized most of the exceeded algorithms. For example, in Katan and Ktantan, where the encoders were using $unint64_t Ptext$. We used to change it to $unint8_t$ and the size collapsed to fit the SRAM and ROM of Arduino UNO, besides the Pi, and also by decreasing the code size for both boards. $ESTATE_T weGIFT128$-128-128, $Romulus$-$M1$-128-128, $Romulus$-$N1$-128-128, $SKINNY$-$AEAD$-$M1$-128-128, $SKINNY$-$AEAD$-$M2$-128-128, $SKINNY$-$AEAD$-$M3$-128-128, $SKINNY$-$AEAD$-$M4$-128-128, $SKINNY$-$AEAD$-$M5$-128-128, and $SKINNY$-$AEAD$-$M6$-128-128 although were minimized but still could not fit the Arduino ROM, and they were eliminated from the Arduino benchmarking only.
2.  Measuring the performance power of the algorithms on the targeted platform requires setting a baseline power for the measured platform in an idle state, i.e., working on no

load or process. After that, the power consumed by the measured algorithm would be the difference between the baseline and the mean measured power.

$$Power_{consumed} = Measured_{power} - Idle_{power} \tag{8}$$

However, during the measuring procedure of the Arduino platform, the resulting algorithms' power started to diminish until they changed to negative values. This problem led to rolling back to check the precision of the baseline power. Thus, the tactic was measuring over time (1 h), from the cool down, the behavior of the platform power in idle mode and then with the load mode (algorithm). The resulting graphs of UNO are shown in Figures 33 and 34 in the two modes.

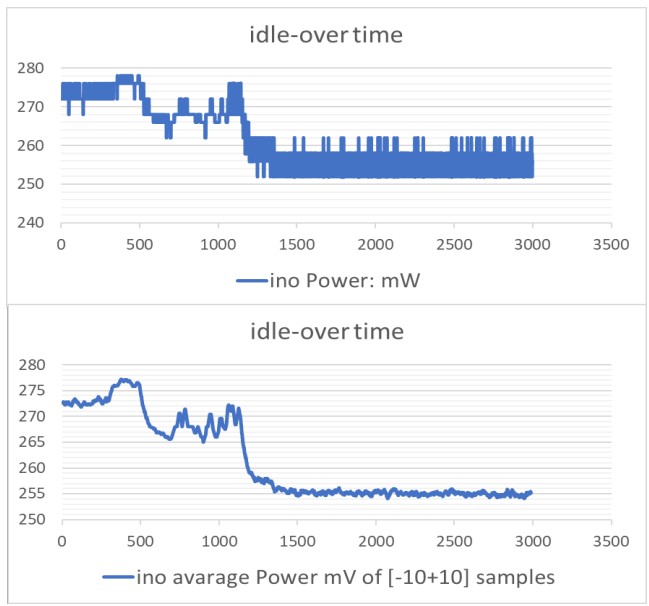

**Figure 33.** 1 h running Arduino in idle mode.

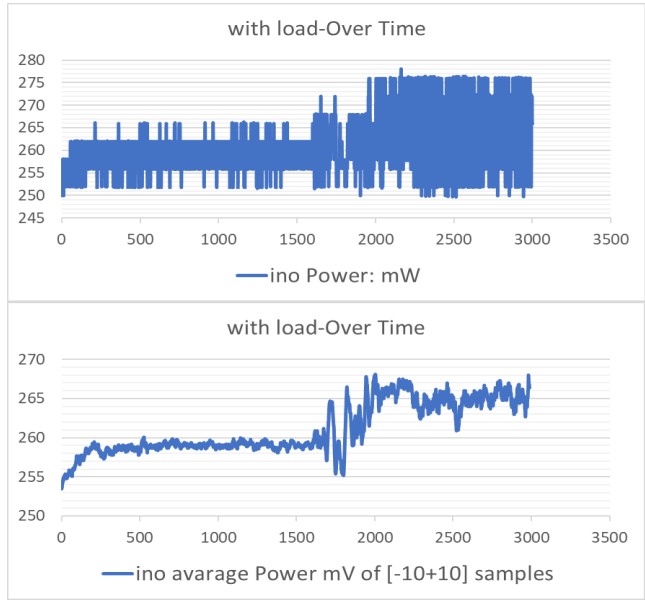

**Figure 34.** 1 h running Arduino in load mode.

The Raspberry Pi has gone through the same process where Figures 35 and 36 reveal the resulting graphs. The graphs show that after 25 min of starting Arduino in the idle mode, a plateau is observed, whereas in the load mode the graph goes to a plateau

after 30 min. Consequently, repeating the measurement by respecting 30 min intervals from the start led to the results interpreted in Section 4, while the Pi graphs show no need for a repeat or respecting any criteria. Furthermore, the speed measurement of UNO was re-benchmarked respecting a 30 min interval.

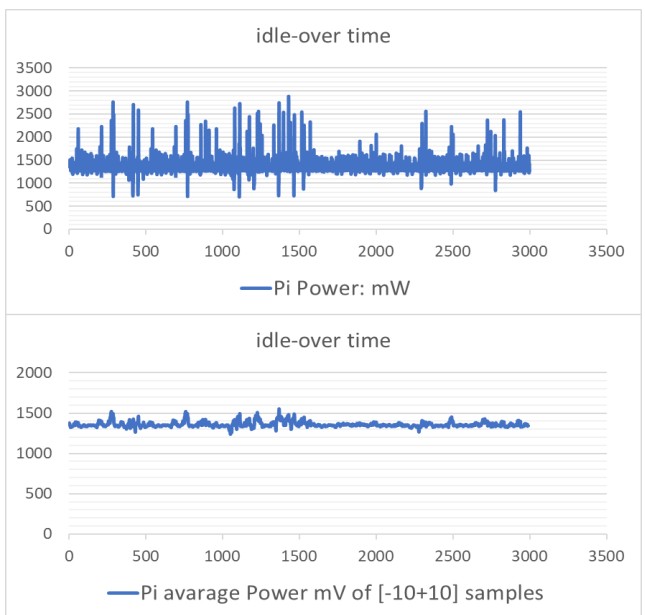

**Figure 35.** 1 h running Pi in idle mode.

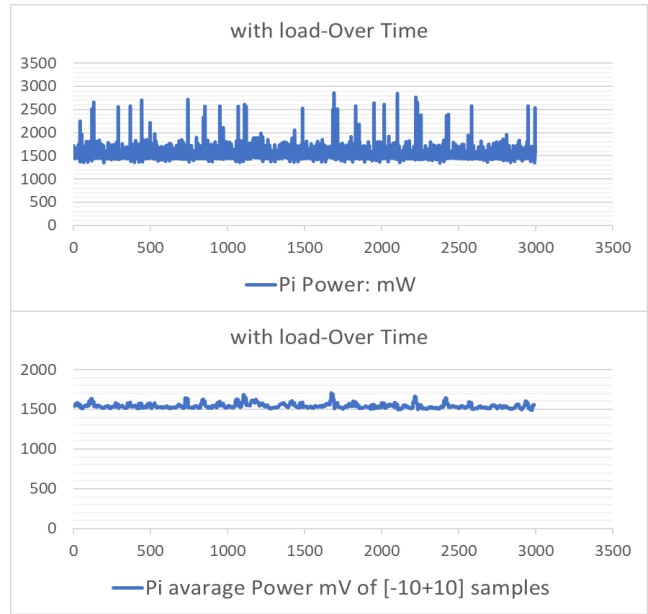

**Figure 36.** 1 h running Pi in load mode.

As a summary, in power, Arduino requires a warm up of about 30 min while Pi does not. The code optimization is essential for Arduino, while for Pi it is not because of the limited memory space in Arduino. In the analysis, an approach of referencing named RR is presented for a comparison between the two targeted platforms. Consequently, if the criteria of comparing a designed LWC algorithm to a standard one like the AES, indicating whether it is LW or not, then according to our work (RR results) it must not depend on the result for only one platform.

## 6. Conclusions and Future Work

Lightweight cryptographic algorithms are important in the IoT because they allow for secure communication on devices with limited processing power and memory. These devices, such as sensors and actuators, often have limited resources and cannot handle the computational demands of traditional cryptographic algorithms. Lightweight algorithms provide a balance between security and performance, making them suitable for use in IoT devices. They are used to secure data transmission, storage, and processing. They play a vital role in ensuring the security of communication and data in IoT-enabled systems. Lightweight cryptographic algorithms are gaining the interest of researchers due to the increasing number of IoT devices being connected to the Internet. IoT devices are becoming more prevalent in a wide range of applications, such as smart homes, industrial automation, and transportation systems. As these devices are often battery powered and have limited processing power, the use of traditional cryptographic algorithms is not always feasible.

In conclusion, lightweight cryptography is challenging work research through the last few years to reach the vision of being lightweight. To our knowledge, what is done in this work is considered unique up to now, by evaluating the performance of the cryptographic algorithms using a variety of metrics and test cases to ensure that the selected algorithms are suitable for use in a wide range of IoT devices. The evaluation process takes into account factors such as the performance on a variety of devices. In this project, a set of 122 ciphers were evaluated and benchmarked using widely used platforms: Arduino and Raspberry Pi. *LEA*-128-128, *COMET*-64_*CHAM*-64-128, *Hight*-64-128, *Speck*-48-72, *Speck*-64-128, and *XTEA*-64-128 were the most promising among the 122 compared algorithms in the power, speed, and ROM measurements. Furthermore, *Schwaemm*-256-128, *GIFT-COFB*-128-128, *Schwaemm*-128-128, *Xoodyak*-128-128, *TinyJAMBU*-192-32-192, and *TinyJAMBU*-128-32-128 are the best performing ciphers among the NIST finalists selected on 29 March 2021. The work/analysis conducted here can be used during the communication occurring between different layers as in the one presented by the authors for a Social IoT architecture [41] and the work conducted by the authors in [42]. As future work or a recommendation, the following aspects can be considered as extensions and enhancements of this work.

1. Arduino mega could be considered in the analysis for the algorithms that exceeded the memory ROM in ATMEGA328P UNO.
2. The NIST finalists were not compared between UNO and Pi with respect to the key schedule, ROM, RAM, and code size, so as a future work these comparisons could be performed.
3. Stream ciphers and hash functions algorithms could be added to the analysis.
4. Using the AES as a relative reference was taken as a linear approach. As future work, it could be established from another approach or approximation after adequate research in such a field and also depending on the behavior of the chosen algorithms.
5. It is important to note that the performance of the algorithm can also be affected by environmental factors, such as the temperature of the Pi/UNO. So, this factor should also be considered while conducting the performance analysis in future works.

**Author Contributions:** M.E.-h. and A.F. conceived of the presented idea. M.E.-h., H.M. and A.F. developed the theoretical analysis. M.E.-h. and A.F. supervised the findings of this work. M.E.-h. took the lead in writing the manuscript with close participation of A.F. All authors provided critical feedback and helped shaping the research, discussed the results and contributed to the final manuscript. All authors have read and agreed to the published version of the manuscript.

**Funding:** The research is funded by the SCS group at the University of Twente.

**Data Availability Statement:** Not applicable.

**Conflicts of Interest:** The authors declare no conflict of interest

## Abbreviations

The following abbreviations are used in this manuscript:

| | |
|---|---|
| LWC | Lightweight Cryptography |
| LW | Lightweight |
| AE | Authenticated Encryption |
| AEAD | Authenticated Encryption with Associated Data |
| AES | Advanced data Encryption Standard |
| ARM | Advanced Reduced Instruction Set Computing Machine |
| ARX | Addition/Rotation/XOR |
| AVR | Alf and Vegard's RISC |
| DEC | Decryption |
| DES | Data Encryption Standard |
| DF | Diffie–Hellman key exchange |
| ENC | Encryption |
| FN | Feistel Networks |
| GFN-2 | Type-2 Generalized Feistel Network |
| GFS | Generalized Feistel Structure |
| HIGHT | High security and Light Weight |
| UNO | Arduino |
| LEA | Lightweight Encryption Algorithm |
| MAC | Message Authentication Code |
| MD5 | Message Digest 5 |
| MSP | Main distribution/Service Panel |
| P | Permutation |
| RAM | Random-Access Memory |
| RFID | Radio Frequency Identification |
| ROM | Read-Only Memory |
| RPi or Pi | Raspberry Pi |
| RR | Relative Reference |
| LD | Linear Dichroism |
| SHA-2 | Secure Hash Algorithm 2 |
| specs | Specifications |
| SPN | Substitution-Permutation Network |
| XTEA | eXtended Tiny Encryption Algorithm |
| IDE | Integrated Development Environment |
| $Ps$ | The size of text in ENC or DEC. |
| $\tau$ | The time taken during one ENC or DEC. |
| $Ks$ | The size of text in expanded key. |
| $f$ | The processor frequency in hertz. |
| $Bs$ | The block size of the algorithm in bytes. |
| $ETh$ | Energy throughput in j/s of ENC and DEC. |
| $Nl$ | Number of loops. |

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
