# Peer review of "Analysis of Lightweight Cryptographic Algorithms on IoT Hardware Platform†"

_futureinternet, doi:10.3390/fi15020054_

Round 1

Reviewer 1 Report

This paper is well written, and the contributions are well presented. Multiple results and metrics justify the superiority of the proposed work. Before considering this paper for publication, the following minor changes must be made.

  1. There are several notations in the paper and it is very challenging to follow them. The authors are summarized using a table for better understanding them.

  2. The conclusion is very long, and there is a lot of general information available. It is recommended to move the general discussion to earlier sections, and to provide a precise conclusion (mainly a summary of results) that focuses on the pitfalls of the proposed work.

  3. Provide a summary of the results at the end of the section. As a result of doing this, the authors can discuss the reasons for the superior performance of the proposed work over existing ones. Furthermore, they can summarize the limitations of the proposed research.

  4. The proposed methodology can be explained through an illustration for better understanding. It is also necessary to measure the computational efficiency of the proposed work.

  5. In the paper, describe the future potential extensions of this work.

  6. List the contributions of this article in the Introduction. Additionally, it is recommended to provide motivation for this project.

  7. The state-of-the-art literature must be presented by considering most recently published works. Recommended to provide some of them from the 'Future Internet' to show the suitability to this journal.

Author Response

1- A table of the notations is added to the paper at the end as abbreviations of notations.

2- The general information is removed from the conclusion, and the conclusion now is providing a summary of the results obtained.

3- A summary of the results with the limitation is added at the end of the discussion section.

4- A future potential extension is added to the conclusion.

5- The contribution and the motivation were added to the introduction.

6- The illustration and the methodology are added before the results section.

7- added a paper published by MDPI in 2022( signals) that talked about LW ( no more was found.

Reviewer 2 Report

This paper presents a survey of Lightweight Cryptography algorithms on IoT.

The topic considered by the survey is interesting and the authors cover it in depth. In the discussion they consider both the more theoretical and the more experimental aspects.

In my opinion, the paper may be very interesting for Future Internet constituents.

I have only two suggestions: 

- the number of papers considered and cited seems small to me for a survey. I suggest that the authors make an effort to identify other papers that are somehow related to the survey theme.

- Innovative architectures have recently been proposed in the context of IoT, for example, Social IoT (see the work of Iera et al.) and Multi-IoT (see the work of Virgili, Cauteruccio, Baldassarre et al.). I suggest that the authors indicate in their conclusions whether and how their analysis can be extended to these two architectures and other innovative IoT architectures that can be found in the literature.

Author Response

1- more papers were added in the background section.

2- linked the results obtained with architectures presented in the literature and how they can benefit from our results.

Round 2

Reviewer 1 Report

The authors are addressed all the comments and may be considered for publication.

Reviewer 2 Report

The authors have tried to comply with my requests. In my opinion, the new version of the paper is better than the previous one and can be published.